# Evaluation of Sleep Quality in a Disaster Evacuee Environment

**DOI:** 10.3390/ijerph17124252

**Published:** 2020-06-15

**Authors:** Hitomi Ogata, Momoko Kayaba, Miki Kaneko, Keiko Ogawa, Ken Kiyono

**Affiliations:** 1Graduate School of Integrated Arts and Sciences, Hiroshima University, Hiroshima 739-8521, Japan; ogawakeicom@hiroshima-u.ac.jp; 2Department of Somnology, Tokyo Medical University, Tokyo 160-0023, Japan; momoko-k@tokyo-med.ac.jp; 3Graduate School of Engineering Science, Osaka University, Osaka 565-8531, Japan; kaneko@bpe.es.osaka-u.ac.jp (M.K.); kiyono@bpe.es.osaka-u.ac.jp (K.K.)

**Keywords:** evacuation shelter, car, sleep, heart rate variability, glucose dynamics

## Abstract

We aimed to evaluate sleep and sleep-related physiological parameters (heart rate variability and glucose dynamics) among evacuees by experimentally recreating the sleep environment of evacuation shelters and cars. Nine healthy young male subjects participated in this study. Two interventions, modeling the sleep environments of evacuation shelters (evacuation shelter trial) and car seats (car trial), were compared with sleep at home (control trial). Physiological data were measured using portable two-channel electroencephalogram and electrooculogram monitoring systems, wearable heart rate sensors, and flash glucose monitors. Wake after sleep onset (WASO) and stage shift were greater in both intervention trials than the control trial, while rapid-eye movement (REM) latency and non-rapid eye movement (NREM) 1 were longer and REM duration was shorter in the evacuation shelter trial than the control trial. Glucose dynamics and power at low frequency (LF.p) of heart rate variability were higher in the car trial than in the control trial. It was confirmed that sleep environment was important to maintain sleep, and affected glucose dynamics and heart rate variability in the experimental situation.

## 1. Introduction

Japan has long suffered from a variety of natural disasters, including strong earthquakes and destructive typhoons, due to its geographical characteristics [1]. In recent years, there has been an increase in the occurrence of typhoons and heavy rain followed by floods/landslides due to climate change, in addition to earthquakes and volcanic eruptions [2]. After a serious disaster, many people are forced to live in public evacuation shelters, such as school gymnasiums and community centers. Based on Japan’s Disaster Relief Act, evacuation shelters are customarily opened as soon as possible and closed within seven days of a serious occurrence; however, they actually take more time to close [3]. For example, evacuation shelters were closed nine months following the Great Hanshin-Awaji Earthquake in 1995 [4]. The environment of those affected by disasters is very harsh. They are forced to sleep directly on the floor without sufficient privacy and air conditioning. Many choose to stay overnight in their car to maintain their privacy or to avoid the crying of babies at night, accompanying pets, etc., despite government recommendations to take shelter in a public place set up by municipalities [5].

Health maintenance following disasters is an important issue in health policy. Factors including decreased access to medical care, reduced physical exercise, and dietary change at evacuation sites are involved in worsening health conditions [6,7]. Although sleep disturbance may be an additional cause, there has been a lack of research on the sleep conditions of victims directly following a disaster. However, many people in evacuation shelters have reported insomnia symptoms, including difficulty initiating sleep and difficulty in maintaining sleep. Additionally, they were unwilling to take sleep medication out of fear of being rendered unable to escape further disasters (e.g., aftershocks) based on the impressions of a healthcare worker who visited evacuation shelters [8]. Regarding the long-term effect of evacuation on sleep, the prevalence of sleep disturbance was 58% (<60 years old) and 68% (≥60 years old) after three weeks [9]. These sleep disturbances are considered an acute stress reaction, insomnia caused by anxiety and/or stress, and so on. According to a retrospective medical chart review study by Kawano et al., crowded shelters had a greater daily incidence of sleep disturbance than non-crowded shelters [10]. In addition, some studies have recreated the sleep environment in evacuation shelters. Mochizuki et al. reported that sleep efficiency and sleeping time decreased significantly when using disaster blankets in the recreated situation of evacuation shelters in winter [11]. Mizuno et al. reported that nocturnal noise levels >40 dB ranged from 53% to 74% in a gymnasium, and the pattern of changes in the noise level was similar to the pattern of subjects’ inability to maintain sleep during the night [12]. Based on these findings, the sleep environment is also an important factor associated with the quality of sleep in an evacuation shelter. Moreover, to the best of our knowledge, there has been no previous study of the quality of sleep while staying overnight in a car.

Sleep disturbance is a risk factor for metabolic syndrome, hypertension, and cerebro-cardiovascular diseases [13]. Insufficient sleep and poor sleep increase sympathetic nervous system activity not only during the night but also during the following day, leading to increased blood pressure [14,15]. This increase in sympathetic nerve activity can trigger the occurrence or aggravation of any of the aforementioned diseases and their related symptoms. Thus, sleep sufficient in both quality and quantity is needed and can be attained by increasing the parasympathetic nervous system during the night. In addition, insufficient sleep and poor sleep are associated with appetite-regulating hormones and glucose metabolism [16,17]. In an experimental study where slow wave sleep (SWS) was selectively disturbed, insulin sensitivity was decreased after three nights of intervention, which suggests that reduced sleep quality may contribute to an increase in the risk of type 2 diabetes [16]. Furthermore, sleep plays an important role in mental health, daytime function, and quality of life [18]. To maintain the physical and mental health of disaster sufferers who spend time in evacuation shelters or in their car following a disaster, an approach in the sleep environment needs to be considered in addition to treatment by healthcare workers as well as an approach for the relief of psychological stress.

To our knowledge, no previous study has investigated sleep and physiological responses (i.e., heart rate variability and glucose dynamics) focusing on the sleep environment of evacuation shelters or cars. Our hypotheses were that the sleep environment of the evacuation shelter and/or car would worsen sleep quality, increase sympathetic nerve activity, and attenuate parasympathetic nerve activity during sleep, and/or insulin sensitivity would decrease and glucose levels rise. Thus, the aim of the present study was to evaluate sleep and sleep-related physiological parameters by experimentally recreating the sleep environment of evacuation shelters and cars.

## 2. Materials and Methods

### 2.1. Subjects

Nine male Japanese young subjects (ages: 21.3 ± 1.3 years; Body Mass Index: 22.0 ± 2.7 kg/m^2^; Pittsburgh Sleep Quality Index score (PSQI): 5.1 ± 2.3; and Morningness-Eveningness Questionnaire score: 46.3 ± 8.8) participated in the present study. Exclusion criteria included food allergies, smoking, chronic diseases, shift worker, planned long-distance jet travel during the study period, self-reported sleep problems, and regular use of medications. This study was approved by the Local Ethics Committee of Hiroshima University, and all subjects provided written informed consent to participate.

### 2.2. Study Design

The experiment was conducted from January to March 2019. We performed two interventions: one assuming the sleep environment of the evacuation shelters (evacuation shelter trial) and another assuming the sleep environment of a car seat (car trial), compared with sleep at home (control trial). To familiarize the subjects with the process of sleeping with a simple electroencephalograph and to confirm the reproducibility of sleep in the control trial, a home trial was performed before both intervention trials (Figure 1, top). We then adopted a second control trial to eliminate the initial effects and confirmed that there were no differences between control trials by pre-analysis. The order of the interventions was alternately assigned. The washout periods were 4–7 days between the first intervention trial and second control trial and 6–11 days between intervention trials to avoid carry-over and seasonal effects. Both interventions were performed in a laboratory set at 20 °C with an air conditioner, and the air conditioner was stopped only during sleeping time. The average room temperature and relative humidity in a laboratory were 18.2 °C and 46.0%, respectively, while those at home were 15.8 °C and 60.0%, respectively. The subjects wore long-sleeved shirts and long pants and performed every trial in the same clothes.

Figure 1 (middle) shows the schedule on the day of the experiment, that is, home and laboratory. Subjects ingested dinner (regular meal: curry rice, 440 kcal, and P:F:C = 8.6:19.2:72.2%) 5 h before going to bed and went to bed at their usual bedtime (23:00 to 3:00) and slept for 8 h. After waking up, they ate breakfast (regular diet: jelly drink, 180 kcal, and P:F:C = 0:0:100%) and remained seated for 3 h. On the day of the experiment, subjects were restricted from caffeine drinks, strenuous exercise, and naps as well as having restricted television and smartphone use from dinner to the end of the experiment the next day.

In the evacuation shelter trial, two cardboard boxes (thickness 8 mm) and one disaster rescue blanket (100% polyester, size 140 × 200 cm) were laid on the laboratory floor, and two disaster rescue blankets were used as comforters. Pillows were brought and used from the home. In the car trial, a car seat with a seat back (backrest) fixed at an angle of 45° from the floor was placed in the laboratory, and two rescue blankets were used as comforters. Subjects took off their shoes and went to bed with their feet down (Figure 1, bottom).

### 2.3. Measurements

#### 2.3.1. Room and Bedclothes Climate

During this period, the ambient temperature and relative humidity of the subject’s bedroom and the climate in the bed (near the chest and near the feet) were recorded every 2 min using a data logger (TR-72Ui, T&D Corp., Matsumoto, Japan).

#### 2.3.2. Assessment of Sleep Quality

Sleep was recorded using portable two-channel electroencephalogram (EEG) and electrooculogram (EOG) monitoring systems (ZA-9, Proassist, Ltd., Osaka, Japan), and sleep stages were manually scored according to the standard criteria [19] by a sleep expert without knowledge of the interventions. Previous studies reported that this portable two-channel device showed strong agreement with polysomnography (PSG); for example, kappa values were 0.80 overall [20] and 0.73–0.86 in the stages of rapid-eye movement (REM), wake, non-rapid eye movement (NREM) 2, and SWS [21], while the agreement with PSG was relatively low (κ = 0.44 [21]; inter-scorer concordance rates were 60.1% in the 2-channel and 71.7% in the PSG [20]) in NREM 1.

#### 2.3.3. Heart Rate Variability

The R-R intervals were measured by electrocardiogram (ECG) signals using wearable heart rate sensors (WHS-1, myBeat, Union Tool Co., Tokyo, Japan). The R-R data were separated consecutively every 5 min and compared with the sleep stage. Following this, if at least one awakening was included in the 5-min note, it was defined as “awake”. If no awakening was included, a stage with a high percentage was adopted. Sleep stages were categorized as NREM sleep or REM sleep. We calculated mean RR, low frequency (LF) (0.04–0.15 Hz; reflects combination of sympathetic nervous system and parasympathetic nervous system influences), high frequency (HF) (0.15–0.4 Hz; reflects parasympathetic nervous system influence), and LF/HF ratio (purported to reflect sympathetic nervous system/parasympathetic nervous system balance) in NREM, REM, and whole sleep. Note that the HF and LF power were divided by total power (HF.p and LF.p).

#### 2.3.4. Continuous Glucose Monitoring

Glucose levels were continuously measured every 15 min using a flash glucose monitor (FreeStyle Libre system; Abott Diabetes Care, Alameda, CA, USA) inserted into the subject’s upper arm. The system is factory calibrated, which eliminates the need for daily calibration during the 14-day wear time. We computed the sleeping glucose level at bedtime as the average 8-h sleeping time, the fasting glucose level as the average 30-min before waking up, and mean indices of meal excursions as glucose rise to peak (ΔG) [22]. We also computed the time course of glucose (glucose dynamics) before sleep (including dinner), every 1 h at sleeping time, and after sleep (including breakfast).

#### 2.3.5. Statistical Analysis

Data are presented as mean values and standard deviations. To identify differences between each intervention trial (the evacuation shelter/car trial) and control trial for room and bedclothes climate, paired *t*-tests were performed. One-way analysis of variance (ANOVA) was used to evaluate the effect of interventions on sleep stages and glucose indices. Two-way ANOVA was used to evaluate the effect of interventions on heart rate variability and glucose dynamics, but these interactions were not considered because they were similar to those of the predictor variables. In the analyses of heart rate variability, trial and sleep stage (REM/NREM) were set as predictor variables. In the analysis of glucose dynamics, trial and sleeping time course (hours) were set as predictor variables. As post-hoc test, multiple comparisons using Dunnett’s test were conducted. The false discovery rate (FDR) was investigated by calculating the *q*-value, that is, FDR analogs of *p* values, with the R package QVALUE [23]. An FDR *q*-value < 0.1 was considered statistically significant. All statistical analyses were performed using R version 3.6.0 (R Foundation for Statistical Computing, Vienna, Austria. http://www.R-project.org/).

## 3. Results

### 3.1. Bed Climate during Sleep

The temperature in the bedclothes near their chest was significantly lower in the evacuation shelter trial (25.0 ± 2.5 °C, *p* < 0.01) and in the car trial (26.8 ± 2.3 °C, *p* < 0.05) than in the control trial (30.1 ± 2.1 °C), while the relative humidity was lower in the evacuation shelter trial (40.0 ± 6.0%, *p* < 0.05) and the car trial (37.2 ± 7.1%, *p* = 0.051) than in the control trial (46.5 ± 7.1%). The temperature in the bedclothes near their feet was lower in the car trial (19.7 ± 2.1 °C, *p* < 0.05) and evacuation shelter trial (20.5 ± 1.9%, *p* = 0.051) than in the control trial (23.2 ± 3.7 °C), while relative humidity in the evacuation shelter trial (42.4 ± 7.3%, *p* < 0.05) and in the car trial (40.4 ± 8.9 °C, *p* < 0.05) were significantly lower than in the control trial (58.2 ± 11.3 °C).

### 3.2. Sleep

One subject was excluded from analysis because in one of the intervention trials, sleep EEG could not be measured successfully due to removal of the electrode. The data of eight subjects who completed three trials were analyzed. The results of sleep architecture in each trial and ANOVA are shown in Table 1. REM latency, wake after sleep onset (WASO), REM duration, NREM 1, and stage shift differed among trials. According to the post hoc test, WASO and stage shift were increased in both the evacuation shelter trial and the car trial compared with the control trial, while REM latency and NREM 1 were longer and REM duration shorter in the evacuation shelter trial than the control trial (Table 2).

### 3.3. Heart Rate Variability

The average values of heart rate variability during sleep are shown in Table 3. Two-way ANOVA showed significant main effects of trials [F(2,36) = 3.604, *p* < 0.05, η_p_^2^ = 0.047, *q* = 0.085], but no significant main effect of sleep stages (REM > NREM) [F(1,36) = 0.258, *p* = 0.615, η_p_^2^ = 0.002] for LF.p. Post hoc tests showed that LF.p was higher in the car trial than in the control trial (Table 2). There were significant main effect of sleep stages (REM > NREM) [F(1,36) = 22.012, *p* < 0.001, η_p_^2^ = 0.036 for mean RR; F(1,36) = 132.995, *p* < 0.001, η_p_^2^ = 0.672 for HF.p; F(1,36) = 21.035, *p* < 0.001, η_p_^2^ = 0.220 for LF/HF ratio] while no significant main effect of trials [F(2,36) = 1.652, *p* = 0.206, η_p_^2^ = 0.005, *q* = 0.238 for mean RR; F(2,36) = 0.258, *p* = 0.774, η_p_^2^ = 0.003, *q* = 0.602 for HF.p; F(2,36) = 2.527, *p* = 0.094, η_p_^2^ = 0.053, *q* = 0.128 for LF/HF ratio].

### 3.4. Glucose Monitoring

The average values of glucose fluctuation from dinner to 3 h after breakfast are shown in Figure 2. Two-way ANOVA showed significant main effects of trials [F(2,250) = 3.226, *p* < 0.05, η_p_^2^ = 0.010, *q* = 0.085] and time course [F(9,250) = 29.851, *p* < 0.001, η_p_^2^ = 0.413]. Post hoc tests showed that glucose dynamics were higher in the car trial than in the control trial (Table 2).

There was no significant difference in the fasting glucose levels and the average value of ΔG among trials (Table 4).

## 4. Discussion

### 4.1. Effect of the Sleep Environment of Evacuees on Sleep

The present study confirmed that an experimentally recreated sleep environment of an evacuation shelter and a car seat can cause disturbed sleep in healthy young adults. The strength of this study was that sleep was evaluated objectively by measuring EEG, in contrast to previous studies that evaluated sleep with an actigraph or questionnaire [11,12]. The results that WASO had increased in the intervention trials were consistent with a previous study finding that WASO increased as measured by actigraph under conditions recreating the sleep environment of an evacuation shelter in winter [11]. In addition to an increase in WASO, we also clarified that light sleep (i.e., NREM 1) and sleep stage shift during sleep increased in the intervention trials. Low agreement between NREM 1 and PSG was found (see Section 2.3.2); however, the increase in NREM 1 shown in the present study was not inconsistent with the increase in WASO. Although the mechanism whereby WASO increased was not determined in this study design, restrictions of body movement, hardness of the bedding, and thermal environment might have affected subjects’ sleep. Our data about the thermal environment showing that the temperature and humidity of the bedclothes were lower in both intervention trials, even though the ambient temperature of the experimental room was higher, supports the possibility that sleep was disturbed by the thermal environment. Further studies, including ones in summer and using heaters in winter, are needed to investigate which factors affect sleep so as to plan countermeasures in the field.

Contrary to our expectations, sleep latency was not prolonged in the intervention trials. The following are possible reasons: (i) the subjects were not experiencing stress and/or anxiety about the disaster in this experiment; and (ii) the subjects were healthy young adults whose desire for sleep was higher than in other age groups. Thus, the sleep of middle-aged and older people, one-third of whom have sleep problems [24], might be more strongly affected than that of the subjects of this study. Although there was no difference in sleep latency, it is easy to imagine that disaster sufferers feel stress and anxiety during evacuation, not only causing difficulty in maintaining sleep, as shown in this study, but also difficulty initiating sleep in a real-world situation. A previous study of the insomnia of disaster victims reported that staff of the nuclear power plant who experienced the Great East Japan Earthquake following the nuclear accident had symptoms of insomnia, especially difficulty initiating sleep, even after three years had passed [25].

While there was no difference in SWS among the intervention and control trials, stage shift had increased in the intervention trials. The subjects had restricted body movement and could not turn over in the driver’s seat during the intervention trials. In the car trial, the seat was set as low as possible in accordance with a previous study, which suggested that the seat should be let down because a smaller angle is more effective for sleep [26]. Further study is needed to investigate the factors of the car trial (i.e., width, angle, or firmness of seat) that decrease SWS. One limitation of the present study is that it considered the thermal environment but not noise. The study of sleep in the gymnasium was measured by actigraph, suggesting that noise delayed sleep onset and increased WASO [12]. A real-world situation would be very harsh due to the fact that an actual evacuation shelter would contain many people, and the level of noise would be much higher.

### 4.2. Effect of the Sleep Environment of Evacuees on the Autonomous Nervous System

Heart rate variability is commonly used as a marker of cardiac autonomic nervous system activity. During NREM sleep, activity of the parasympathetic nerves was predominant, inducing a decrease in heart rate. During REM sleep, there is a large fluctuation in heart rate due to autonomic nervous system irregularity [27,28]. While there was no significant difference between the intervention and control trials, differences in heart rate variability between NREM and REM sleep were confirmed in the present study. Compared with the control trial, LF power was significantly higher in the car trial. This suggests that sympathetic nervous activity usually increased during the sleep periods when parasympathetic nerve activity was dominant, indicating insufficient rest. This is consistent with the reduction in SWS in the car trial. That is, even when compared with NREM sleep (except for waking), sleep was shallow, affecting sympathetic nervous activity, in the car trial. According to previous studies, sympathetic nervous activity continued to be enhanced due to poor sleep until the day after waking up. For example, short sleep increases blood pressure not only at night but also during the daytime [7,13,14]. Patients with obstructive sleep apnea syndrome experience recurrent episodes of upper airway collapse during sleep, causing apneas and hypopneas that result in arousal from sleep. Sympathetic nervous system activity is thus increased at night and parasympathetic nervous system activity decreases during the daytime [15]. Increased sympathetic nervous activity leads to the onset of increased blood pressure and cardiovascular diseases such as heart disease and cerebrovascular disease. It is important to ensure sufficient sleep quality and quantity at night because the parasympathetic nervous system becomes dominant at the point when enough rest is obtained. It is necessary to examine mechanisms such as the angle of the seat and the effect of turning over in the car trial, and make interventions to reduce the burden on evacuees in the future. Although the experiment involved healthy young subjects in the present study, the possibility of strong stress or anxiety from a disaster is also a factor, as well as the fact that the effect might be even worse in middle-aged and/or elderly people. Further studies of middle-aged and elderly people are needed in the future.

### 4.3. Glucose Dynamics

There were no differences between the control trials and intervention trials in the fasting glucose levels and average value of ΔG, while glucose dynamics were higher in the car trial than in the control trial. An association between sleeping time and blood glucose levels has been reported in previous studies. An animal study showed that sleep deprivation of just one day may alter liver function involved in blood glucose and insulin control, increasing the risk of developing type 2 diabetes [29]. A human study suggested that sufficient sleep duration improved fasting blood glucose levels and insulin secretion in healthy young adults [30]. In another study that excluded SWS selectively for three consecutive days, the magnitude of the decrease in insulin sensitivity was strongly correlated with the magnitude of the reduction in SWS on the third day [16], suggesting the possibility that long-term sleep disruption might change glucose dynamics. Moreover, overnight intervention in the present study might have altered glucose dynamics due to the worsened sleep situation. In actual evacuation life, sleep restrictions caused by the sleep environment in the shelters or in cars would continue long-term, possibly worsening glucose dynamics. Further studies to investigate the long-term effect of sleep in evacuation shelters or in a car on glucose dynamics are needed.

Sleep is also associated with hunger hormones; insufficient sleep results in a decrease in leptin levels (appetite-suppressing hormone) and an increase in ghrelin levels (appetite-enhancing hormone) the following day [17]. In the present study, subjects were provided with fixed meals without regard for their appetite because the situation was modeled on evacuation conditions. If the subjects could eat ad libitum, the amount of energy intake and following glucose dynamics might thus differ across trials. It is necessary to consider eating hormone-related parameters, including subjective hunger and fullness, amount of energy intake, and meal contents, in future studies.

### 4.4. Limitations

This study has some limitations. First, although the hardness and narrowness of the bedding could have reproduced the conditions of sleeping in an evacuation shelter or a car, there is no denying that the temperature and humidity might have been quite different from those in an actual gymnasium or car. Moreover, we could not limit the movements of the legs and body position changes in the car trial or induce claustrophobia. Second, we targeted healthy young men in the present study, but the subjects might have included individuals with sleep problems (e.g., insomnia, sleep apnea, and sleep–wake phase delay), as suggested by their PSQI scores and bedtimes. Further studies targeting women and the elderly are also needed. Third, the sample size of the present study was small; thus, the results might have been underestimated due to a lack of statistical power.

## 5. Conclusions

Sleep environments in an evacuation shelter and car were reproduced experimentally in the present study to compare them with sleep at home. Additionally, the effects on sleep, autonomic nervous activity, and glucose dynamics were examined. As a result, WASO and stage shift were found to have increased in both intervention trials compared with the control trial, while REM latency and NREM 1 were longer and REM duration was shorter in the evacuation shelter trial than the control trial. Glucose dynamics and power at low frequency (LF.p) of heart rate variability were higher in the car trial than in the control trial. It was confirmed that sleep environment was important to maintaining sleep and affected glucose dynamics and heart rate variability in the experimental situation. Improvements in the sleep environment under evacuation are needed that might contribute to the prevention of cardiovascular diseases and disaster-related deaths.

## Figures and Tables

**Figure 1 ijerph-17-04252-f001:**
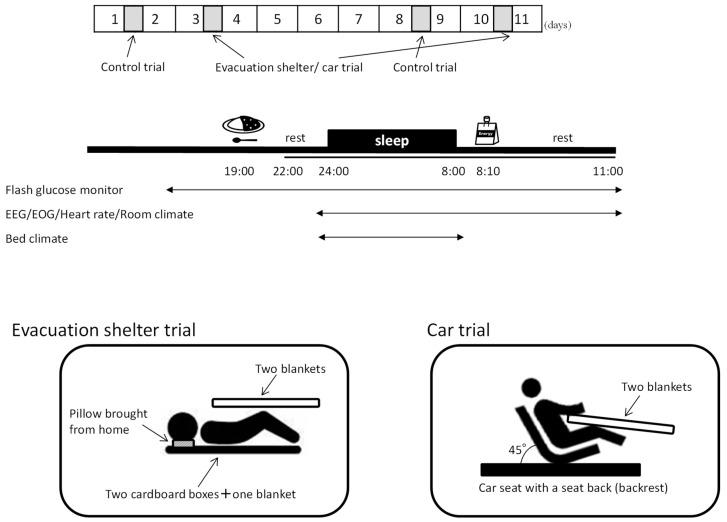
Study protocol: schematic overview of the study protocol (**top**); time schedule of sleep interventions for subjects who always go to sleep at 24:00 (**middle**); and pattern diagram for each trial (**bottom**). All subjects ate the same meals; dinner was curry and rice; breakfast was a jelly drink. On the day of the experiment, subjects were restricted from caffeine drinks, alcohol consumption, strenuous exercise, and naps, and had restrictions on television and smartphone use from dinner to the end of the experiment the next day.

**Figure 2 ijerph-17-04252-f002:**
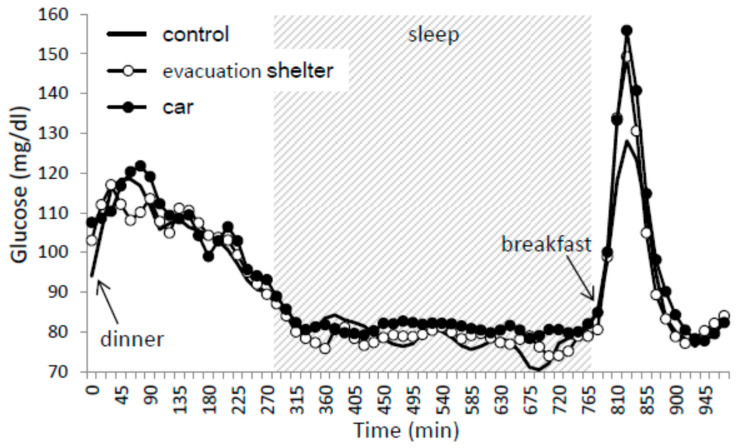
Time course of glucose from dinner to 3 h after breakfast. Mean values of blood glucose for nine subjects were plotted every 15 min. The black lines show data for the control trial, the open circles show data for evacuation shelter trials, and the closed circles show data for the car trial. Sleep hours are indicated in the gray zone.

**Table 1 ijerph-17-04252-t001:** Results of sleep stages using the portable two-channel electroencephalogram monitoring system (*n* = 8).

Trial	Control	Evacuation Shelter	Car	ANOVA				FDR *q* Value
	Mean (95% CI)	Mean (95% CI)	Mean (95% CI)	df	F	*p*	η_p_^2^	
Sleep latency	18.2	15.1	8.8	(2,15)	0.148	0.864	0.010	0.602
(min)	(6.4–30.0)	(4.8–25.5)	(1.4–16.1)					
REM latency	72.7	116.8	144.3	(2,15)	3.710	**0.049**	0.229	**0.089**
(min)	(42.5–102.8)	(78.4–155.1)	(95.6–193.0)					
WASO	1.3	6.0	7.4	(2,15)	8.841	**0.003**	0.369	**0.014**
(%)	(0.1–2.4)	(2.4–9.6)	(3.0–11.7)					
REM	26.9	23.4	18.8	(2,15)	14.269	**0.000**	0.412	**0.005**
(%)	(23.1–30.7)	(18.8–28.0)	(16.9–20.8)					
NREM 1	3.0	4.2	5.0	(2,15)	3.712	**0.049**	0.212	**0.089**
(%)	(0.9–5.2)	(3.1–5.3)	(3.2–6.7)					
NREM 2	50.5	48.0	53.6	(2,15)	2.843	0.090	0.185	0.128
(%)	(46.1–54.9)	(44.6–51.5)	(50.6–56.6)					
SWS	18.4	18.4	15.3	(2,15)	3.002	0.080	0.117	0.128
(%)	(14.8–21.9)	(14.6–22.2)	(12.2–18.4)					
Stage shift	46.6	68.5	81.1	(2,15)	11.423	**0.001**	0.351	**0.007**
(times)	(33.7–59.6)	(58.6–78.4)	(64.9–97.4)					

REM, rapid eye movement; WASO, wake after sleep onset; NREM, non-rapid eye movement; SWS, sleep wave sleep; ANOVA, analysis of variance; FDR, false discovery rate; CI, confidence interval; percent divided by total sleep time. The bold numbers show significant difference.

**Table 2 ijerph-17-04252-t002:** Dunnett’s post hoc test.

Comparisons			Mean Difference	SE	*p*	95% CI	FDR *q* Value
Sleep stage	REM latency	Control vs. Evacuation shelter	58.778	21.830	**0.031**	5.5–112.1	**0.081**
		Control vs. Car	37.639	22.830	0.206	−18.1–93.4	0.238
	WASO	Control vs. Evacuation shelter	6.367	1.576	**0.002**	2.5–10.2	**0.012**
		Control vs. Car	4.871	1.648	**0.018**	0.8–8.9	**0.075**
	REM	Control vs. Evacuation shelter	−7.967	1.495	**0.000**	−11.6–−4.3	**0.005**
		Control vs. Car	−3.446	1.564	0.078	−7.3–0.4	0.128
	NREM 1	Control vs. Evacuation shelter	2.389	0.880	**0.030**	0.2–4.5	**0.081**
		Control vs. Car	1.382	0.920	0.260	−0.9–3.6	0.279
	Stage shift	Control vs. Evacuation shelter	38.000	8.020	**0.001**	18.4–57.6	**0.005**
		Control vs. Car	23.625	8.387	**0.024**	3.1–44.1	**0.077**
Heart rate variability	LF.p	Control vs. Evacuation shelter	2.025	1.116	0.136	−0.5–4.6	0.171
		Control vs. Car	2.843	1.065	**0.021**	0.4–5.3	**0.075**
Glucose	Dynamics	Control vs. Evacuation shelter	0.324	1.408	0.962	−2.8–3.5	0.602
		Control vs. Car	3.246	1.408	**0.041**	0.1–6.4	**0.085**

REM, rapid-eye movement; WASO, wake after sleep onset; NREM, non-rapid eye movement; LF.p, low frequency divided by its total power ; CI: confidence interval; FDR: false discovery rate. The bold numbers show significant difference.

**Table 3 ijerph-17-04252-t003:** Results of heart rate variability during NREM and REM sleep (*n* = 7).

Sleep-Stage	NREM			REM			NREM + REM		
Trial	Control	Evacuation Shelter	Car	Control	Evacuation Shelter	Car	Control	Evacuation Shelter	Car
	Mean	Mean	Mean	Mean	Mean	Mean	Mean	Mean	Mean
	(95% CI)	(95% CI)	(95% CI)	(95% CI)	(95% CI)	(95% CI)	(95% CI)	(95% CI)	(95% CI)
Mean RR	1155.8	1095.5	1146.6	1092.9	1039.5	1086.4	1139.9	1087.0	1134.3
(ms)	(1045–1266)	(953–1238)	(1007–1286)	(990–1196)	(924–155)	(952–1221)	(1033–1247)	(956–1218)	(997–1272)
HF.p	28.7	28.5	29.7	12.5	12.3	13.3	24.4	24.7	26.3
(%)	(24.1–33.3)	(22.3–34.8)	(23.2–36.1)	(10.5–14.4)	(8.2–16.3)	(10.7–15.9)	(20.2–28.6)	(19.2–30.2)	(21.1–31.4)
LF.p	22.4	24.8	26.2	23.1	24.5	24.6	22.5	24.5	25.8
(%)	(18.0–26.8)	(20.6–29.1)	(21.1–31.2)	(19.6–26.7)	(19.9–29.1)	(20.5–28.6)	(18.5–26.5)	(20.4–28.6)	(21.1–30.5)
LF/HF	1.41	1.87	1.62	2.30	3.35	2.45	1.65	2.27	1.78
ratio	(1.0–1.8)	(1.0–2.7)	(1.1–2.2)	(1.7–2.9)	(2.1–4.6)	(1.8–3.1)	(1.2–2.1)	(1.2–3.4)	(1.3–2.3)

NREM: non-rapid eye movement; REM: rapid eye movement; HF.p: high frequency divided by its total power; LF.p: low frequency divided by its total power; LF/HF ratio: low frequency/high frequency ratio; CI: confidence interval.

**Table 4 ijerph-17-04252-t004:** Results of glucose indices (*n* = 9).

Trial	Control	Evacuation Shelter	Car	ANOVA				FDR *q* Value
	Mean (95% CI)	Mean (95% CI)	Mean (95% CI)	df	F	*p*	η_p_^2^	
Fasting glucose level	82.3	78.6	83.0	(2,16)	1.208	0.325	0.046	0.323
(mg/dL)	(78.2–86.5)	(70.3–86.8)	(74.4–91.6)					
ΔG	67.4	74.3	76.1	(2,16)	0.972	0.400	0.041	0.361
(mg/dL)	(52.2–82.6)	(61.4–87.3)	(60.1–92.1)					

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
