# Peer review of "Evaluation of Sleep Quality in a Disaster Evacuee Environment"

_ijerph, 2020, doi:10.3390/ijerph17124252_

Round 1

Reviewer 1 Report

accept as it

Reviewer 2 Report

The authors addressed in a satisfactory fashion all of my concerns. The study seems now more accurate from a statistical point of view, perfectly reliable for the quality/exposition of the results and clearer for what concerns the explanations of the methods. Well done.

As a final suggestion, I recommend using nu as that is the most frequently used acronym for normalized units (e.g. LF.nu).

This manuscript is a resubmission of an earlier submission. The following is a list of the peer review reports and author responses from that submission.

Round 1

Reviewer 1 Report

Thank you for the opportunity to review this manuscript. In it, the authors present the findings of a small experiment examining the simulated effects of sleeping in atypical environments (e.g., disaster shelter, car) following a disaster. The findings suggest that, while sleep architecture is compromised relative to a control condition, there are limited other physiological impacts (within the constraints of the experimental protocol).

Overall, I find the study to be reasonably well contextualized and discussed. Additionally, there is, generally, an appropriate acknowledgment of the limitations of the study. However, I have the following comments/recommendations:

  1. Given that the methodological purpose of the manuscript is to examine sleep, please consider reducing the focus at the beginning of the introduction on other detrimental factors (i.e. DVTs). Upon initial reading, the first paragraph suggested that DVTs were going to be the focus of the manuscript rather than sleep (as the title indicated). Obviously, this switched by the midpart of the second paragraph, but there is a dissonance between the paragraphs.

  2. Given that null hypothesis testing was employed, please report the a priori hypotheses for this research.

  3. Please provide a rationale for the sample size. Was this determined a priori or simply a convenience sample. If driven by a power analysis, please provide sufficient details to independently verify this.
  4. The exclusionary criteria do not indicate any sleep-related conditions or shift-work as exclusionary criteria. These conditions may impact the well-controlled nature of the study (though certainly not the real-world generalizability). Was any attempt beyond the PSQI made to ensure that these individuals were "typical" sleepers prior to enrollment? If not why not and please comment on the impact this may have had. 

  5. Relatedly, the PSQI scores and "usual bedtimes" reported suggest potential insomnia and circadian phase delays within the group. This should be discussed somewhere.

  6. On line 107, there is a "2-7" day washout period. Why was this not consistent? For whom was it inconsistent? Please provide a more substantive rationale for this.

  7. Please provide information about the validity/reliability of the 2-channel EEG relative to laboratory-grade PSG. 

  8. Why were comparisons between the shelter and car not performed? Part of the context of the study is that individuals choose to sleep in the car to mitigate environmental issues in the shelter. Thus the lack of comparison between these two seems curious.

  9. Given that the sleep parameters are not entirely independent of each other (i.e., sleep efficiency and WASO are directly related), p-values for comparisons across these measure should be adjusted to control family-wise error rates (i.e., Bonferroni, Holm, Sidak, etc). The same is true for the heart rate variables. I am ok if these are presented as both uncorrected and corrected. 
  10. Effect sizes (i.e. Cohen's dz, Hedges g) + 95% confidence intervals should be reported for pairwise comparisons.

  11. The discussion should appropriately contextualize any edits made from the points above.

Regardless, this is important work that can help to inform future, larger studies and I look forward to reading a revised version.

Author Response

This document has been edited by professional editors at Editage, a division of Cactus Communications.

  1. Given that the methodological purpose of the manuscript is to examine sleep, please consider reducing the focus at the beginning of the introduction on other detrimental factors (i.e. DVTs). Upon initial reading, the first paragraph suggested that DVTs were going to be the focus of the manuscript rather than sleep (as the title indicated). Obviously, this switched by the midpart of the second paragraph, but there is a dissonance between the paragraphs.

We thank you for these helpful comments. We have revised the Introduction section as you suggested (Lines 41–61).

  1. Given that null hypothesis testing was employed, please report the a priori hypotheses for this research.

We have added the hypotheses in the Introduction section (Lines 79–81).

  1. Please provide a rationale for the sample size. Was this determined a priori or simply a convenience sample. If driven by a power analysis, please provide sufficient details to independently verify this.

The sample size was set empirically in accordance with our previous studies that evaluated sleep using a cross-over design (Sleep Med. Res. 2019, 10(2), 67–74; Environ. Health Prev. Med. 2014, 19(5), 354–361). In addition, it was difficult to add more data for fear of including seasonal effects. However, we agree with your comments that we should provide a rationale for the sample size. In this study, we discussed the defect of small sample size as a study limitation (Lines 392–393). In future studies, we would asset the sample size using a power analysis, as you pointed out.  

  1. The exclusionary criteria do not indicate any sleep-related conditions or shift-work as exclusionary criteria. These conditions may impact the well-controlled nature of the study (though certainly not the real-world generalizability). Was any attempt beyond the PSQI made to ensure that these individuals were "typical" sleepers prior to enrollment? If not why not and please comment on the impact this may have had.

The exclusion criteria were revised (Lines 88–90), and the characteristics of the subjects indicated by the PSQI scores were discussed (Lines 390–391).

  1. Relatedly, the PSQI scores and "usual bedtimes" reported suggest potential insomnia and circadian phase delays within the group. This should be discussed somewhere.

We added this problem as a study limitation (Lines 390–391).

  1. On line 107, there is a "2-7" day washout period. Why was this not consistent? For whom was it inconsistent? Please provide a more substantive rationale for this.

The description of the washout period was misleading. In the present study, control trials were set before each intervention trial. The control trials were the same as usual sleep at home, assuming no carry-over effect. The interval between the first intervention trial and the second control trial was 4–7 days, and that between intervention trials was 6–11 days. The washout period was set at 4 days or more to avoid a carry-over effect within 2 weeks to avoid seasonal effects. The lack of a carry-over effect of the first intervention was also confirmed by a comparison of control trials.

To clarify this, we have revised the text (Lines 101–103).

  1. Please provide information about the validity/reliability of the 2-channel EEG relative to laboratory-grade PSG.

We added this information in the Methods section (Lines 138–142). As we described, 2-channel EEG showed good agreement with PSG in the sleep stages, except for stage N1.

  1. Why were comparisons between the shelter and car not performed? Part of the context of the study is that individuals choose to sleep in the car to mitigate environmental issues in the shelter. Thus the lack of comparison between these two seems curious.

In the future, comparisons between these environments should be considered, as you pointed out. In the present study, we aimed to investigate the burden in the disaster evacuees’ sleep environments. Thus, we used Dunnett’s multiple comparison to compare the shelter/car with the usual situation.

  1. Given that the sleep parameters are not entirely independent of each other (i.e., sleep efficiency and WASO are directly related), p-values for comparisons across these measure should be adjusted to control family-wise error rates (i.e., Bonferroni, Holm, Sidak, etc). The same is true for the heart rate variables. I am ok if these are presented as both uncorrected and corrected.

Thank you for your very insightful comment. We did not consider family-wise error in the earlier manuscript. We agree with your comment that sleep efficiency and WASO are directly related, so we omitted the parameter sleep efficiency.” Similarly, “RMSSD” was omitted from the results of the heart rate variables. We added the q-value, with which the false discovery rate (FDR) was investigated (Lines 171–173, Table 1, Table 2, Lines 245–252, Lines 274–275, and Table 4).

  1. Effect sizes (i.e., Cohen's dz, Hedges g) + 95% confidence intervals should be reported for pairwise comparisons.

Effect sizes (ηp2) and 95% confidence intervals have been added (Table 1, Table 2, Lines 245–252, Table 3, Lines 274–275, and Table 4).

  1. The discussion should appropriately contextualize any edits made from the points above.

We revised the Discussion section according to the revisions (Lines 296–393).

Reviewer 2 Report

The presented paper assesses a disaster evacuees' sleep environment and effects on physiological parameters.

The paper is well written.

Although the present data are intriguing, there are several issues that should be addressed:

  1. Abstract is not so well structured
  2. As a one more limitation of study can be mentioned small number of patients

Author Response

  1. Abstract is not so well structured

Thank you for your helpful comments.

We have revised the Abstract section in accordance with your comment (Lines 12–24).

  1. As a one more limitation of study can be mentioned small number of patients

We added a small sample size as a study limitation (Lines 392–393).

Reviewer 3 Report

In this well presented and designed study, Ogata and coworkers use an experimental model to investigate the variations of sleep parameters and cardio-metabolic responses to sleeping into evacuees’ environments, showing how such condition brings to actual, slight sleep disruption and impaired sympatho-vagal balance. However of interest, especially for the country it comes from, this work presents some concerns that need to be addressed.

  1. The temperature topic was too vague. Generally speaking, it is not clear why the authors investigated it. It seems out of the scope of the study and, although important from a practical point of view, does not allow the authors to draw any definite conclusions. Assuming that the laboratory was indoor, it is not clear why the (average? Specify) temperature (and, accordingly, the humidity) should have dropped more for the shelter arm than the car arm. Did the time of the year (January to March) influenced these variations? Why was the control night not set with a standard room temperature and kept constant through the night (comparable to normal “at home” conditions)? What was the reason why the temperature was recorded at feet and chest levels? And finally, the authors stated that the air conditioning was switched off before putting the patient to sleep (at lights out?): if this is the case, a more optimal room temperature could have favored the sleep onset and justify the normal sleep latency values in the results.
  2. For the statistical analysis, ANOVA seems to be a better statistical fit.
  3. In the results, the authors present very low values of REM sleep latency during the control arm compared to the others, but still not significant. Please revise this, also according to point 2. Moreover, sleep efficiency is reduced in the shelter and car arms, but to a level that is still considered on the high end of normality. Generalizing conclusions on this topic only judging from the particular response of a selected group of people (who are also particularly less exposed to sleep efficiency drops) seems too embryonal.
  4. It would be good if the authors could report also VLF power in the heart rate variability section. On this note, they should try to normalize LF and HF for the total power.

I also have a number of minor concerns:

  1. The mean PSQI indicated in the study would suggest that the quality of sleep of the participants was already somewhat altered. Please discuss.
  2. In the car arm it is not clear to me how the movements of the individuals were limited. Normally, people would have the steering wheel, the glove department or the front seats limiting the movements of the legs and the position changings, however this did not seem like it was reproduced in the experimental model. Another thing to take into account (to be mentioned in the limitations) is claustrophobia, that some people would suffer of, especially in real, stressful conditions, and that would somewhat impact sleep quality/quantity.
  3. The authors mention that the control arm was performed at home, but this does not seem to be the case, as the exam was described as an in-lab procedure.
  4. Did the authors exclude sleep disorders when recruiting participants? Were OSA and PLMs investigated? Was alcohol intake assessed prior to the study commencement?
  5. Please specify that HF power reflects mainly parasympathetic activity.
  6. Figure 1, right bottom panel: typo, brought.
  7. Line 176: typo, intervention trial.
  8. Line 180: REM duration. Table 1: specify the meanings of the % in brackets.

Author Response

This document has been edited by professional editors at Editage, a division of Cactus Communications.

  1. The temperature topic was too vague. Generally speaking, it is not clear why the authors investigated it. It seems out of the scope of the study and, although important from a practical point of view, does not allow the authors to draw any definite conclusions. Assuming that the laboratory was indoor, it is not clear why the (average? Specify) temperature (and, accordingly, the humidity) should have dropped more for the shelter arm than the car arm. Did the time of the year (January to March) influenced these variations? Why was the control night not set with a standard room temperature and kept constant through the night (comparable to normal “at home” conditions)? What was the reason why the temperature was recorded at feet and chest levels? And finally, the authors stated that the air conditioning was switched off before putting the patient to sleep (at lights out?): if this is the case, a more optimal room temperature could have favored the sleep onset and justify the normal sleep latency values in the results.

Thank you for your helpful comments.

The description of room temperature and relative humidity has been moved from the Results section to the Methods section (Lines 104–106) as a characteristic of the laboratory. Ambient temperature and relative humidity were almost the same between the shelter trial and car trial, so we obtained these data together. In the Discussion section, “4.1. The thermal environment in the experiments was deleted. The results of bed climate have been retained in the Results section as additional information because the thermal environment sometimes affects sleep and should be considered. To clarify the address that the reviewer pointed out, we have revised the Discussion section (Lines 309–313). Regarding the final question, we recorded the temperature at the chest and feet, referring to a previous study [Tsuzuki et al., Int. J. Biometeorol. 2018, 52(4), 261–270]. 

  1. For the statistical analysis, ANOVA seems to be a better statistical fit.

We reanalyzed the data using ANOVA (Lines 165–171, Table 1, Table 2, Lines 244–252, Table 3, Lines 274–275, and Table 4).

  1. In the results, the authors present very low values of REM sleep latency during the control arm compared to the others, but still not significant. Please revise this, also according to point 2. Moreover, sleep efficiency is reduced in the shelter and car arms, but to a level that is still considered on the high end of normality. Generalizing conclusions on this topic only judging from the particular response of a selected group of people (who are also particularly less exposed to sleep efficiency drops) seems too embryonal.

According to Point 2 above, we have reanalyzed the data using ANOVA (Lines 165–171, Table 1, Table 2, Lines 244–252, Table 3, Lines 274–275, and Table 4). Sleep efficiency was omitted (please see #1–9).

  1. It would be good if the authors could report also VLF power in the heart rate variability section. On this note, they should try to normalize LF and HF for the total power.

Thank you for your helpful comments. We reanalyzed the power of high frequency (HF.p) and low frequency (LF.p) components divided by the total power (Lines 152–153).

[Minor concerns]

  1. The mean PSQI indicated in the study would suggest that the quality of sleep of the participants was already somewhat altered. Please discuss.

Please see #1–5.

  1. In the car arm it is not clear to me how the movements of the individuals were limited. Normally, people would have the steering wheel, the glove department or the front seats limiting the movements of the legs and the position changings, however this did not seem like it was reproduced in the experimental model. Another thing to take into account (to be mentioned in the limitations) is claustrophobia, that some people would suffer of, especially in real, stressful conditions, and that would somewhat impact sleep quality/quantity.

Thank you for your helpful comments.

We added this problem as a study limitation (Lines 385–389).

  1. The authors mention that the control arm was performed at home, but this does not seem to be the case, as the exam was described as an in-lab procedure.

In the control trials, the subjects slept at home under the schedule of the protocol. We have added this information to the manuscript (Lines 108–109).

  1. Did the authors exclude sleep disorders when recruiting participants? Were OSA and PLMs investigated? Was alcohol intake assessed prior to the study commencement?

Individuals with sleep disorders were excluded only by self–assessment information. As you pointed out, the possibility of OSA and PLMs cannot be denied. We have added this issue as a study limitation (Lines 389–391). Regarding alcohol intake, the subjects were restricted, and we have added this point to the manuscript (Lines 126–127).

  1. Please specify that HF power reflects mainly parasympathetic activity.

We have revised the manuscript (Lines 150–151).

  1. Figure 1, right bottom panel: typo, brought.

We have revised the manuscript (Figure 1).

  1. Line 176: typo, intervention trial

We have revised the manuscript (Line 187).

  1. Line 180: REM duration. Table 1: specify the meanings of the % in brackets.

We have revised the manuscript (Lines 190 and 221).